# Automated Generation of Hospital Discharge Summaries Using Clinical Guidelines and Large Language Models

**Simon Ellershaw**[1], **Christopher Tomlinson**[1,2,3], **Oliver Burton**[4], **Thomas Frost**[1], **John Gerrard Hanrahan**[4,5], **Danyal Z Khan** [4,5], **Hugo Layard Horsfall**[4,5], **Mollie Little**[4], **Evaleen Malgapo**[1], **Joachim Starup-Hansen**[4], **Jack Ross**[4], **George Woodward**[4], **Martinique Vella-Baldacchino**[6], **Kawsar Noor**[1,2,3], **Anoop D Shah**[1,2,3] **Richard JB Dobson**[1,2,3,7]

[1]Institute of Health Informatics, University College London, London, United Kingdom
[2]National Institute for Health and Care Research Biomedical Research Centre, University College London Hospitals National Health Service Foundation Trust, London, United Kingdom
[3]Health Data Research UK, London, United Kingdom
[4]University College London Hospitals NHS Foundation Trust, London, UK
[5]Wellcome/EPSRC Centre for Interventional and Surgical Sciences, University College London, London, UK
[6]MSK Lab, Imperial College London, London, UK
[7]Department of Biostatistics and Health Informatics, King's College London, London, UK
Corresponding Author: simon.ellershaw.20@ucl.ac.uk

## Abstract

Discharge summaries are essential yet time-consuming documents doctors write at the end of a patient's hospital stay. They are the primary form of communication between hospital and community care teams. The automatic generation of summaries could reduce the administrative burden on doctors. We propose to use large language models, few-shot prompted by clinical guidance, to perform this task. Unlike previous supervised approaches, our method does not require a large training dataset, can accept full-length physician notes as inputs and is explicitly guided by clinical best practice. We implemented such a system using Royal College of Physicians London guidelines, GPT-4-turbo and MIMIC-III physician notes. 53 summaries were evaluated by 11 clinicians and found to have a micro accuracy of 0.81. Finally, we discuss methodical limitations and the required future improvements to the evaluation framework.

## Introduction

A clinician must write a discharge summary at the end of every patient's hospital stay. The summary communicates to the post-hospital care team what has happened to the patient during their hospital stay and their ongoing care plan (Kind and Smith 2008). However, this manual process adds to clinicians' workloads and can be of varying quality (Rattray et al. 2017).

Therefore, the automation of this process using machine learning models has been proposed as a solution (Patel and Lam 2023). Current state-of-the-art approaches (Pal et al. 2023) fine-tune encoder-decoder models (Lewis et al. 2019) to map a set of clinician notes to a discharge summary. However, this supervised approach faces challenges due to the limited training data, extended length of clinician notes and variable ground truth quality (Searle et al. 2023).

Recently, the scaling of the training and size of natural language auto-regressive transformers has led to a new class of models known as large language models (LLMs) (Brown et al. 2020). LLMs have shown the ability to learn from a few examples, accept inputs over 100,000 words and attain state-of-the-art performance on several benchmark tasks, including text summarisation (Liang et al. 2022; Anthropic 2023). Such model properties could solve several problems currently faced in the automatic generation of discharge summaries.

This work presents the first LLM-based discharge summary generator to be tested on full clinical notes and evaluated by clinicians. Our key contribution is the use of clinical guidelines to prompt the LLM with the desired format and content of a summary instead of learning this from the data.

## Methodology

We converted guidelines from the UK's Royal College of Physicians London (RCP) (Royal College of Physicians 2021), see Fig 2, to a JSON schema. We excluded the medication section, which requires the non-trivial merging of structured e-prescribing data with the extraction of the reasons for any medication changes from the clinical notes.

Following this, we created a fixed prompt of a system message containing the JSON schema and a one-shot example generated from an exemplar RCP discharge summary (Royal College of Physicians 2021). For full details of this process see Appendix 2.

To test the efficacy of the method, we used the freely-available MIMIC-III v1.4 dataset (Johnson et al. 2016; Johnson, Pollard, and Mark 2016; Goldberger et al. 2000). We filtered the notes table for hospital admissions for which a discharge summary exists and so could be generated and at least one physician note. Next, we removed extraneous characters, artefacts from the anonymity process and the notes were deduplicated by keeping only the first occurrence of a

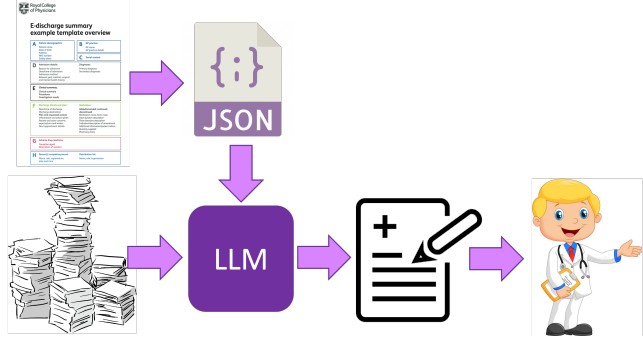

Figure 1: Shows the proposed method, which combines discharge summary guidelines and physician notes into an LLM prompt in order to produce a discharge summary for review by a clinician.

line of text.

For our experiments, we used GPT-4-turbo version 1106-Preview (OpenAI 2023a), with temperature=0, due to its strong benchmark performance (Liang et al. 2022) and 128k context window, which allowed all sets of tested physician notes to be accepted in a single query.

One round of qualitative evaluation was performed with a clinician using a sample of 5 hospital admissions. We used this feedback to adjust the description of a select number of fields. For a complete list see Table 2.

We evaluated the final system using a team of 11 UK-qualified doctors and physician associates with prior experience writing discharge summaries. After reading the physician notes and clinical guidelines, the clinicians were asked to evaluate the number of times the following errors occurred for each discharge summary field: missed severe, missed minor, additional hallucination and additional not relevant. A missed error was categorised as severe if it had the potential to meet the NHS England (NHS England National Patient Safety Team 2023) definition of medium to severe levels of harm. Each clinician evaluated five summaries, of which one was duplicated with another clinician to allow the calculation of inter-annotator agreement.

## Results

53 discharge summaries were generated and evaluated. The median input physician notes length after de-duplication was 4996 tokens and the fixed prompt was 5057 tokens, measured using the cl100k_base tokeniser (OpenAI 2021). The median inference time was 40.59s at a median API cost of $0.12. The model extracted 25.07% of the generated elements verbatim from the input physician notes. For a further breakdown of these metrics, see Table 3.

We found the median number of errors per summary to be 7, with the error proportions to be 36.28% missed severe, 27.44% missed minor, 14.55% added hallucination and 21.73% added not relevant. One summary failed to conform to the JSON schema. We calculated the percentage agreement between annotators, see Eqn 8, to be 59.72%.

To calculate the performance metrics in Table 1, we used

| Section | Recall | Precision | Acc |
|---|---|---|---|
| Admission Details | 0.90 | 0.95 | 0.85 |
| Allergies And Adverse Reaction | 0.98 | 1.00 | 0.98 |
| Clinical Summary | 0.76 | 0.92 | 0.71 |
| Diagnoses | 0.84 | 0.94 | 0.80 |
| Discharge Details | 0.93 | 0.96 | 0.89 |
| Patient Demographics | 1.00 | 0.84 | 0.84 |
| Plan And Requested Actions | 0.90 | 0.88 | 0.80 |
| Social Context | 0.96 | 0.88 | 0.84 |
| Macro Average | 0.91 | 0.92 | 0.84 |
| Micro Average | 0.86 | 0.92 | 0.81 |

Table 1: Recall, precision and accuracy metrics per section for discharge summaries generated from MIMIC-III notes as evaluated by clinicians.

Equations 1-7, defining a missed error as a false negative and an addition error as a false positive. Table 4 shows a per-field view of the same results. The GP Practice section is excluded from analysis, as the GP is not a role in the American healthcare system and so the section was never filled.

## Discussion

While the metrics in Table 1 show promise for many fields, safety-critical errors, such as missed severe and hallucinations, highlight the challenges in using LLMs for discharge summarisation and the need for clinician-in-the-loop review at the point of use. However, this in turn poses the risk of automation bias arising over time

The evaluation of this work was limited to a single centre's ICU data due to data-availability, in scale due to the labour-intensive nature of clinical evaluation and the low inter-annotator agreement metric shows the variability of clinical review for this task. Therefore, the development of a clinical grounded, scalable and systematically repeatable evaluation framework is vital future work.

The key strength of this work is that, to the author's knowledge, it is the first to show the effectiveness of using clinical guidelines to prompt LLMs for administrative medical tasks, such as discharge summarisation. This overcomes the main limitations of supervised approaches, namely the need for large labelled datasets and the inherent biases encoded in training on real-world data of variable quality.

## Conclusion

This work proposes a method to generate draft hospital discharge summaries using clinical guidelines to prompt LLMs. Unlike supervised training, this requires only a single training example and explicitly follows current best practices. A team of clinicians evaluated such a system using GPT-4-turbo, RCP guidelines and physician notes from MIMIC-III to have a micro accuracy of 0.81. However, further development of the evaluation framework is required for the improvement and safe deployment to clinical practice of such a method.

## Ethical Considerations and Reproducibility Statement

Access to the MIMIC-III dataset requires an approval process, including mandatory data ethics training. All authors, including clinical evaluators, undertook this process.

The PhysioNet Credentialed Data Use Agreement (PhysionNet 2023a), which governs the use of the MIMIC-III dataset, explicitly prohibits sharing access to the data with third parties. Therefore, in line with MIMIC's guidance on the use of third-party LLMs (PhysionNet 2023b) all GPT-4 queries were made using Azure OpenAI service whilst being opted out of the human review of the data.

Concerning reproducibility, we cannot openly share the generated summaries and evaluations due to the terms of the MIMIC dataset license. However, the code to produce the summaries is open-sourced (https://github.com/simonEllershaw/llm-discharge-summaries), allowing a MIMIC-credentialed user to reproduce the summaries evaluated in this work. Similarly, all data analysis scripts are also released.

## Ethics Board Approval

The collection of patient information and creation of the MIMIC-III research resource was previously reviewed by the Institutional Review Board at the Beth Israel Deaconess Medical Center, which granted a waiver of informed consent and approved the data-sharing initiative (Johnson et al. 2016). No additional specific ethics board approval was required for this project.

## Acknowledgments

SE is supported by a UCL UKRI Centre for Doctoral Training in AI-enabled Healthcare studentship (EP/S021612/1). ADS is supported by research grants from EPSRC (EP/Y018087) and NIHR (AI_AWARD01864). CT is supported by a UCL UKRI Centre for Doctoral Training in AI-enabled Healthcare studentship (EP/S021612/1), a MRC Clinical Top-Up, a studentship from the NIHR Biomedical Research Centre at University College London Hospital NHS Trust, and the Health Data Research UK Phenomics and Prognostic Atlas Theme.

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

# Appendix 1- Royal College Of Physician Guidelines

- The discharge summary should be brief, containing only pertinent information on the hospital episode, rather than duplicating information which GPs already have access
- Below describes a template for a generic discharge summary, created for the purposes of this learning activity and will not be identical to the form used within your organisation, where you may find slightly different content or other terms being used.
- The template is based on the standard for e-discharge summaries, published by the Professional Record Standards Body and available online: https://theprsb.org/standards/edischargesummary/
- * Several of the elements will contain information which aligns with clinical coding. This will be done by using drop-down lists in your organisation's system or by software identifying terminology which can be coded in the background - this means it is very important to use terms accurately and appropriately. Marked *

| Section | Headings and elements | Notes |
|---|---|---|
| A | **Patient demographics** | Check the correct patient record is being completed, especially where autopopulated by the electronic patient record |
| | Patient name | Autopopulated |
| | Date of birth | Autopopulated |
| | Patient address | Autopopulated |
| | NHS number | Autopopulated (unique identifier) |
| | Safety alerts: | Any alerts could be documented here eg treatment limitation decisions, multi-resistant organisms, refusal of specific managements eg blood products; safeguarding concerns. This includes risks to self (eg suicide, overdose, self-harm, neglect), to others (to carers, professionals or others) and risks from others (risk from an identified person eg family member). |
| B | **GP practice** | |
| | GP name | Name of a patient's general practitioner, if offered by the patient or their representative |
| | GP practice details | Autopopulated - Name and address of the patient's registered GP practice |
| C | **Social context** | Includes elements such as lifestyle factors eg smoking status, alcohol, and social context, eg whether the person lives alone. This is particularly important if the admission and discharge locations differ. Consider what information a new carer would need to know. More detailed information would be recorded in forms, such as "This is me" form for dementia patients. Also includes educational history. |
| D | **Admission details** | |
| | Reason for admission* | The main reason why the patient was admitted to hospital, eg chest pain, breathlessness, collapse, etc. |
| | Date/time of admission | Autopopulated |
| | Admission method | May be autopopulated, eg elective/emergency |
| | Relevant past medical, surgical and mental health history | Whilst the GP is likely to hold this information it is useful for documents to stand-alone and provides an insight into the basis for clinical decisions. Includes relevant previous diagnoses, problems and issues, procedures, investigations, specific anaesthesia issues, etc |
| | **Diagnoses** | List / bullet points/ brief factual information |
| | Primary diagnosis* | *Confirmed* primary diagnosis (or symptoms); active diagnosis being treated. Record to highest level of certainty, eg do not record a diagnosis if it is not certain, record a symptom instead. |
| | Secondary diagnoses* | Record any other diagnoses relevant to admission, such as: other conditions which impact on the treatment eg dementia, diabetes, COPD; complications during admission eg venous thromboembolism, hospital acquired pneumonia; or incidental new diagnoses. |
| E | **Clinical summary** | |
| | Clinical summary | Details of the patient's journey can be written in this section, including details about the patient's admission and response to treatments, recorded as a summary narrative. Very concise, where possible. |
| | Procedures* | The details of any therapeutic or diagnostic procedures performed. This should be the name of the procedure, with additional comments if needed. |
| | Investigation results | It is important to include results of investigations which the GP is likely to monitor either of the health condition or associated with medication use eg renal function in patients with diabetes or prescribed an ACE inhibitor. This is also an opportunity to provide more detail on medical problems not related to the main admission eg current lung function tests in patient with COPD admission for elective procedure; cardiac echogram, etc |
| F | **Discharge details and Plan** | It is really important the GP understands the next steps for the patient and what they are responsible for organising |
| | Date/time of discharge | Autopopulated |
| | Discharge destination | Highlight when different to patient's usual address and if permanent or interim arrangement eg residential care, rehabilitation facility, local hospital (from tertiary centre) |
| | **Plan and requested actions:** | Make clear where the responsibility for actions lies (eg with the GP practice or hospital). eg Health or test monitoring, specialist services eg Macmillan, Diabetes, Optometry |
| | Information and advice given | Note of information and advice given and patient/carer comprehension |
| | Patient and carer concerns, expectations and wishes | Description of the concerns, wishes or goals of the person in relation to their care, as expressed by the person, their representative or carer. Also record who has expressed these. Where the person lacks capacity this may include their representative's concerns, expectations or wishes. |
| | Next appointment details | Follow-up appointment booked, eg outpatient department - include contact details. |
| | **Medication** | All information required to prescribe medication, quantity supplied, pharmacy check |

| Medication name* | Form* | Route* | Dose duration description* | Dose directions description | Indication*/ description of any amendment | Additional Information/ patient advice | Quantity supplied | Pharmacy check |
|---|---|---|---|---|---|---|---|---|
| May be generic name or brand name | Form of the medicinal substance eg capsules, oral, tablets, liquid | Medication administration description (eg oral, intravenous, etc). May include method (eg inhaler). | Recommendation of time period for which the medication should be continued. eg "Continue indefinitely"; "Do not discontinue" (never discontinue), "Stop when course complete". | Description of entire medication dosage and administration directions, including dose quantity and medication frequency; eg "1 tablet at night" or "20mg at 10pm" | Reason for medication being prescribed, where known. Description of any amendment, where relevant | May include guidance to prescriber, patient or person administering the medication eg rinse mouth with water after use | The quantity of the medication eg tablets, (eg tablets, inhalers, etc.) provided to the patient on discharge. This may be dispensed by the pharmacy or on the ward. Or "Patient's own medication". | Initials of pharmacist |

| Section | Headings and elements | Notes |
|---|---|---|
| | Status: Added/amended | |
| | Status: Continued | |
| | Status: Discontinued (also to include date of discontinuation) | |
| G | **Allergies and adverse reactions** | "No known drug allergies or adverse reactions" should be recorded where a specific agent is not mentioned |
| | Causative agent* | The agent such as food, drug or substances that has caused or may cause an allergy intolerance or adverse reaction in this patient. |
| | Description of reaction* | A description of the manifestation of the allergic reaction experienced by the patient. Eg skin rash. |
| H | **Person completing record** | Autopopulated; multiple authors could contribute to discharge summary eg ward doctor, pharmacy, therapists, nursing staff, but this the individual clinician who is responsible for completing the discharge summary. |
| | Name | Role | Organisation | Date and time completed | Additional information |
| | **Distribution list** (cc and to include patient) | May be automated depending on electronic record used; print copy for patient and go through it with them to check for accuracy and ensure understanding. A copy of the discharge summary should be sent to the admission referrer where relevant, in addition to the GP. |
| | Name | Role | Organisation |

Figure 2: A copy of the RCP crib sheet outlining their guidelines for discharge summary writing (Royal College of Physicians 2021).

# Appendix 2-LLM Prompt

To form the LLM prompt, firstly, we take guidelines written by the RCP, see Fig 2, (Royal College of Physicians 2021) and using the title and description of each section convert

this to a JSON schema shown in Listing 1. We excluded the medication section, which requires the non-trivial merging of structured e-prescribing data with the extraction of the reasons for any medication changes from the clinical notes. The schema's required and title fields are redundant and removed to reduce input length.

Next, we convert an exemplar discharge summary from the RCP guidelines to JSON according to the schema. The accompanying physician notes are formatted and de-duplicated using the same method as outlined in the methodology sections for the MIMIC physician notes. Together, the RCP JSON schema, one-shot prompt and set of input physician notes form the input prompt, as shown in Fig 3.

Figure 3: The GPT-4-turbo (OpenAI 2023a) prompt used in this work. Contained in bold braces are the variables produced by the processes outlined in the methodology section. System, user and assistant refer to the different roles used by OpenAI's chat completions API (OpenAI 2023b).

Listing 1: RCP-based discharge summary JSON schema. For presentation purposes, only the patient_demographics section is shown.

```
1   {
2     "description": "The discharge summary
         should be brief, containing only
         pertinent information on the
         hospital episode, rather than
         duplicating information which GPs
         already have access to in their own
         records.",
3     "type": "object",
4     "properties": {
5       "patient_demographics": {
6         "$ref": "#/definitions/
           PatientDemographics"
7       },
8       ...
9     }
10    "definitions": {
11      "AdmissionDetails": {
12        "type": "object",
13        "properties": {
14          "reason_for_admission": {
15            "description": "The main
               reason why the patient was
               admitted to hospital, eg
               chest pain, breathlessness,
               collapse, etc. This should
               be symptoms and not the
               diagnosis.",
16            "type": "string"
17          },
18          "admission_method": {
19            "description": "Eg elective/
               emergency",
20            "type": "string"
21          },
22          "relevant_..._history": {
23            "description": "Whilst the GP
               is likely to hold this
               information it is useful
               for documents to stand-
               alone and provides an
               insight into the basis for
               clinical decisions.
               Includes relevant previous
               diagnoses, problems and
               issues, procedures,
               investigations, specific
               anaesthesia issues, etc",
24            "type": "array",
25            "items": {
26              "type": "string"
27            }
28          }
29        }
30      },
31      ...
32    }
33  }
```

Table 2 shows the alterations to the prompt descriptions after 1 round of qualitative clinical evaluation.

## Appendix 3- Metric Equations

In order to calculate the performance metrics shown in Tables 1 and 4, we first defined the evaluation of each field as a 4-dimensional vector (sum missing severe errors, sum missing minor errors, sum additional hallucination errors, sum additional not relevant errors).

From this definition we calculated the number of additional errors for a given field $f$ summed across all generated summaries as the number of false positives, $FP_f$ and likewise for missing errors and false negatives $FN_f$. The number of positive predictions for a field, $P_f$, is defined as either the length of list type fields or the number of sentences for string type fields. Therefore, the number of true positives, $TP_f$, for a field $f$ is

$$TP_f = P_f - FP_f \qquad (1)$$

From this and given that true negatives do not exist in this framework, the field's precision, $p_f$, recall, $r_f$, F1, $F1_f$ and accuracy, $acc_f$ scores, can be calculated,

$$p_f = \frac{TP_f}{TP_f + FP_f}, \qquad (2)$$

$$r_f = \frac{TP_f}{TP_f + FN_f}, \qquad (3)$$

$$F1_f = 2 \times \frac{p_f \times r_f}{p_f + r_f}, \qquad (4)$$

$$acc_f = \frac{TP_f}{TP_f + FP_f + FN_f}. \qquad (5)$$

We found the average precision scores by averaging across all fields

$$p_{macro} = \frac{1}{|p|} \sum_f p_f. \qquad (6)$$

Or by first pooling across fields

$$p_{micro} = \frac{\sum_f TP_f}{\sum_f TP_f + \sum_f FP_f}. \qquad (7)$$

Similar equations hold for averaging recall, F1 and accuracy.

To calculate the inter-annotator agreement for the set of all doubly evaluated field, $f$, we defined two 2-D vector $(FN_{f1}, FP_{f1})$ and $(FN_{f2}, FP_{f2})$ one for each evaluator. $FN$ and $FP$ were chosen as they are the evaluation defined inputs to Eqn 7. $A_o$ was then calculated as

$$A_o = \frac{\sum_f \delta\{(FN_{f1}, FP_{f1}), (FN_{f1}, FP_{f1})\}}{|f|} \qquad (8)$$

where the $\delta$ function is defined as

$$\delta_{a,b} = \begin{cases} 1, a = b \\ 0, a \neq b. \end{cases} \qquad (9)$$

## Appendix 4- Additional Results

| Section | Field | Change to Description |
|---------|-------|----------------------|
| Admission Details | Reason For Admission | Added- "This should be symptoms and not the diagnosis." |
| | Admission Method | Remove- "May be autopopulated" |
| Diagnoses | Secondary Diagnoses | Added- "Do not include diagnoses made before this hospital admission." |
| Clinical Summary | Procedures | Added- "Do not include procedures performed before this hospital admission." |
| | Investigation Results | Added- ", chest x-ray, mri scan, etc. Each investigation is a separate element in the list." |
| PlanAndRequestedActions | Post Discharge Plan and Requested Actions | Added- Do not include jobs that are still to be done in hospital before discharge." |
| | Next Appointment Details | Added- "Note date and contact details if available." |

Table 2: A table showing the alterations made to the field descriptions of the RCP discharge summary JSON schema after 1 round of clinical evaluation.

| | Percentile | | | |
|---|---|---|---|---|
| | 25th | 50th | 75th | Max |
| De-Duplicated Physician Note Length / Tokens | 2793 | 4996. | 8772 | 95682 |
| Output Note Length / Tokens | 705 | 807 | 884 | 1234 |
| Inference Time / secs | 33.41 | 40.60 | 48.61 | 125.95 |
| Inference Cost / $ | 0.10 | 0.12 | 0.16 | 1.04 |

Table 3: Table of system properties when tested on MIMIC-III notes. The fixed prompt length is 5057 tokens. We calculated token lengths using cl100k_base tokenizer (OpenAI 2021)

.

| Section | Field | Mean Number of Elements | Proportion of Blank Values | Recall | Precision | F1 | Acc |
|---|---|---|---|---|---|---|---|
| Admission Details | Admission Method | 1.00 | 0.00 | 0.93 | 0.96 | 0.94 | 0.89 |
| | Reason For Admission | 1.00 | 0.00 | 0.79 | 0.92 | 0.85 | 0.74 |
| | Relevant Past Medical And Mental Health History | 8.34 | 0.08 | 0.91 | 0.95 | 0.93 | 0.87 |
| Allergies And Adverse Reaction | Causative Agent | 1.87 | 0.00 | 0.98 | 1.00 | 0.99 | 0.98 |
| | Description Of Reaction | 1.87 | 0.09 | 0.98 | 1.00 | 0.99 | 0.98 |
| Clinical Summary | Clinical Summary | 4.28 | 0.00 | 0.71 | 0.98 | 0.82 | 0.70 |
| | Investigation Results | 4.30 | 0.04 | 0.75 | 0.86 | 0.80 | 0.67 |
| | Procedures | 2.36 | 0.28 | 0.87 | 0.94 | 0.91 | 0.83 |
| Diagnoses | Primary Diagnosis | 1.00 | 0.00 | 0.83 | 0.94 | 0.88 | 0.79 |
| | Secondary Diagnoses | 3.45 | 0.13 | 0.84 | 0.94 | 0.89 | 0.80 |
| Discharge Details | Discharge Destination | 1.00 | 0.00 | 0.93 | 0.96 | 0.94 | 0.89 |
| Patient Demographics | Safety Alerts | 1.74 | 0.72 | 1.00 | 0.84 | 0.91 | 0.84 |
| Plan And Requested Actions | Information And Advice Given | 1.40 | 0.55 | 0.98 | 0.80 | 0.88 | 0.79 |
| | Next Appointment Details | 1.00 | 0.72 | 1.00 | 0.89 | 0.94 | 0.89 |
| | Patient And Carer Concerns Expectations And Wishes | 1.25 | 0.62 | 0.89 | 0.83 | 0.86 | 0.75 |
| | Post Discharge Plan And Requested Actions | 7.89 | 0.00 | 0.88 | 0.90 | 0.89 | 0.80 |
| Social Context | Social Context | 2.89 | 0.17 | 0.96 | 0.88 | 0.91 | 0.84 |
| Macro Average | | | | 0.90 | 0.92 | 0.90 | 0.83 |
| Micro Average | | | | 0.86 | 0.92 | 0.89 | 0.81 |

Table 4: Evaluation metrics per discharge summary field, including mean number of elements and proportion of blank values per field as well as recall, precision, F1 and accuracy.