# OpenReview forum: "Automated Generation of Hospital Discharge Summaries Using Clinical Guidelines and Large Language Models"
_AAAI.org/2024/Spring_Symposium_Series/Clinical_FMs — AAAI 2024 SSS on Clinical FMs_

### Official Review · Reviewer_JoP1 · 2024-02-16

**Rating:** 7
**Confidence:** 4

**Review:**

Quality and clarity:
- The writing is generally clear and easy to understand
- Methods seem to be clear and well-described for reproducibility

Originality and significance:
- Physician-guided LLM prompting seems original and highly relevant for this workshop
- Prompt examples in the appendix are quite interesting, as well as the performance breakdown per discharge summary field

Weaknesses:
- Figure 1: There is a black box around the cartoon doctor, please remove
- It would be interesting to see what the standard deviation would be under multiple generations of the same discharge summary, as the performance of chatgpt may change over time, but I understand the expensive nature of human-labeling
- 53 summaries were evaluated by 11 clinicians seems small. Also, do clinicians agree with each other in their analysis?

---

### Official Review · Reviewer_GjCn · 2024-02-22
**Review of Automated Generation of Hospital Discharge Summaries Using Clinical Guidelines and LLMs**

**Rating:** 6
**Confidence:** 3

**Review:**

The paper, "Automated Generation of Hospital Discharge Summaries Using Clinical Guidelines and Large Language Models," explores an approach to automating the creation of hospital discharge summaries. Leveraging LLMs few-shot prompted by clinical guidelines, this method does not require extensive training datasets and can handle full-length physician notes (by utilizing large LLMs), guided by clinical best practices. The system, tested with GPT-4-turbo and MIMIC-III physician notes, was evaluated by clinicians, achieving a micro accuracy of 0.81. The paper discusses methodological limitations and future improvements needed in the evaluation framework.

Pros:
1. Utilizes LLMs in conjunction with clinical guidelines to automate the creation of discharge summaries, offering a solution to reduce manual effort.
2. The methodology does not rely on extensive datasets for training, potentially making it more adaptable and easier to implement across different hospital systems.
3. Involves clinicians in the evaluation process, ensuring that the automated summaries meet practical clinical needs and standards.

Cons:
1. The reliance on a single dataset (MIMIC-III) for testing may limit the generalizability of the findings to other healthcare environments or patient demographics.
2. Accuracy and Completeness: While achieving a micro accuracy of 0.81 is promising, there may still be concerns about the accuracy and completeness of the generated summaries, especially in complex cases. A more in-depth error analysis of identifying which group of patients (which health conditions, which demographics, etc) end up with a better model performance can significantly improve the work.
3. Even though the paper acknowledges limitations in its evaluation framework, suggesting further refinement and broader testing is necessary to fully assess the system's effectiveness and reliability.

---

### Official Review · Reviewer_4B96 · 2024-02-22
**A decent naive approach to generating discharge summaries with clinician evaluation**

**Rating:** 8
**Confidence:** 5

**Review:**

Summary
-- the study generated discharge summaries using GPT-4 based on a MIMIC-III dataset (with secure Azure instance). The quality of the summaries was evaluated by clinicians.

Pros:
-- good use of secure Azure instance in accordance with Physionet DUA
-- rigorous clinician evaluation with clear labeling criteria (NHS harm definitions) and duplication for inter-rater agreement
-- good use of json format template with initial experimentation to adjust prompts. Good use of one-shot in-context example.
-- good reporting of inference time and costs, in addition to performance

Cons:
-- small dataset, n=53
-- low but realistic inter-rater agreement at 60%
-- could have used JSON-enforcing
-- A better definition for the performance metrics would be an atomic-fact approach to evaluating the accuracy of statements in the summary. Metrics here could be fact precision and recall. Using the total number of sentences could inflate the denominator and thus deflate the "FPs" and "FNs"
-- this is a relatively naive approach to discharge summary synthesis (dumping all the prior notes in-context). A more compelling study could have included a comparison to RAG-based approaches.

---

### Official Review · Reviewer_qX16 · 2024-02-23
**Sound evaluation of LLM application in hospital discharge summaries generation**

**Rating:** 9
**Confidence:** 4

**Review:**

The paper applied LLMs to automate the generation of hospital discharge summaries from physician notes, and evaluated the performance with clinicians from different clinical perspectives. Public benchmark dataset MIMIC-III was used.
- Quality: the paper is technically sound and of high quality. Data processing and filtering criteria, such as only including the first occurrence of duplicate text, is considerable to avoid potential information leakage.
- Clarity: the paper is well-structured, precise and concise in experiment settings, results interpretation and limitation.
- Originality: the paper illustrated a novel application of existing technologies to a challenging and meaningful clinical task. Though the idea might not be the first, the implementation and thorough evaluation with clinicians make the paper sufficiently novel.
- Significance: the clinical task of discharge summary is a challenging and important problem to address using advanced ML techniques. Moreover, the evaluation criteria of different types of mistakes in particular with LLM applications could be a good reference for other similar clinical applications using LLMs.

Questions and feedback:
1. How many clinicians reviewed the same discharge summary? If there's more than 1 reviews per summary, how you aggregate the evaluation metrics, especially how to handle conflict opinions.
2. It will be better to compare the current approach with some baselines to further improve the technical soundess, e.g. fine-tune encoder-decoder models from supervised training.